# Structure and Inhibition of the Human Na^+^/H^+^ Exchanger SLC9B2

**DOI:** 10.3390/ijms26094221

**Published:** 2025-04-29

**Authors:** Sukkyeong Jung, Surabhi Kokane, Hang Li, So Iwata, Norimichi Nomura, David Drew

**Affiliations:** 1Department of Biochemistry and Biophysics, Science for Life Laboratory, Stockholm University, 171-65 Stockholm, Sweden; sukkyeong.jung@dbb.su.se (S.J.); surabhi.kokane@dbb.su.se (S.K.); hang.li@scilifelab.se (H.L.); 2Graduate School of Medicine, Kyoto University, Yoshida Konoe-cho, Sakyo-ku, Kyoto 606-8501, Japan; iwata.so.2z@kyoto-u.ac.jp (S.I.); nnomura@mfour.med.kyoto-u.ac.jp (N.N.)

**Keywords:** membrane transporters, NHE, cryo-EM, lipid remodeling, SLC, cell volume regulation, RBC, hypertension

## Abstract

The sodium/proton exchanger NHA2, also known as SLC9B2, is important for insulin secretion, renal blood pressure regulation, and electrolyte retention. Recent structures of bison NHA2 has revealed its unique 14-transmembrane helix architecture, which is different from SLC9A/NHE members made up from 13-TM helices. Sodium/proton exchangers are functional homodimers, and the additional N-terminal helix in NHA2 was found to alter homodimer assembly. Here, we present the cryo-electron microscopy structures of apo human NHA2 in complex with a Fab fragment and also with the inhibitor phloretin bound at 2.8 and 2.9 Å resolution, respectively. We show how phosphatidic acid (PA) lipids bind to the homodimer interface of NHA2 on the extracellular side, which we propose has a regulatory role linked to cell volume regulation. The ion binding site of human NHA2 has a salt bridge interaction between the ion binding aspartate D278 and R432, an interaction previously broken in the bison NHA2 structure, and these differences suggest a possible ion coupling mechanism. Lastly, the human NHA2 structure in complex with phloretin offers a template for structure-guided drug design, potentially leading to the development of more selective and potent NHA2 inhibitors.

## 1. Introduction

The transmembrane exchange of protons (H^+^) for sodium (Na^+^) ions by Na^+^/H^+^ exchangers (NHEs) is a fundamental mechanism that is thought to be carried out in every cell [1,2,3]. In mammals, there are nine different SLC9A/NHE members, which differ in their tissue distribution, subcellular localization, kinetics, and regulation [2,3,4]. NHE1 to NHE9 are well-known for their roles in human physiology, such as Na^+^ reabsorption in the kidney and acid–base homeostasis [2,5,6,7]. The mammalian NHA1 and NHA2 members belong to the SLC9B family, which share greater sequence similarity with bacterial Na^+^/H^+^ exchangers, e.g., NapA [8,9] and have only been more recently identified [4,10,11,12]. Based on its tissue expression pattern, genomic location, and phloretin sensitivity, NHA2/SLC9B2 activity has been linked to the Na^+^(Li^+^) counter-transport activity associated with the development of essential hypertension and diabetes in humans [11,13,14,15,16]. Consistent with this interpretation, NHA2 aids in sodium reabsorption in the kidney [17,18] and is a critical component of the with-no-lysine kinase 4-sodium-chloride cotransporter (WNK4-NCC) pathway in the regulation of blood pressure [18]. Importantly, both in vitro and in vivo studies show that NHA2 contributes to β-cell insulin secretion and blood–glucose homeostasis [19]. Thus, NHA2 has an important role in both metabolism and renal physiology and understanding its transport mechanism and regulation is of utmost importance.

Na^+^/H^+^ exchangers are physiological homodimers, and structural studies have shown that each protomer is made up of a dimer (scaffold) domain and a core (ion translocation) domain [9,20,21,22]. Na^+^/H^+^ exchangers operate by an “elevator” alternating access mechanism [9,21,22,23,24], wherein ions are transported by the core domain against the dimerization domain, which remains fixed due to oligomerization [24]. The structures of mammalian SLC9A and SLC9B members, as well as those from bacterial homologues, have shown that the architecture of the 6-TM core ion translocation domain remains remarkably well-conserved [9,21,22,25,26,27]. The largest structural divergence is in the dimerization domain [21]. The clearest example is NHA2 [8], which has an additional domain-swapped helix at the N-terminus (TM−1), which in the absence of lipids, forms a weak dimeric interface. The binding of crude phosphatidylinositol (PI) lipids to the dimerization interface of NHA2 was shown to stabilize a more extensive homodimer by readjusting the position of the N-terminal TM−1 helix [8]. Thus, it is plausible that the binding of specific lipids between the protomers in NHA2 may act to switch on activity via the lipid-dependent stabilization of the homodimer.

While the functional consequences of the lipid-induced remodeling of NHA2 are unclear, there is a precedent for specific lipids regulating the activity of Na^+^/H^+^ exchangers by binding to their dimerization interfaces. For example, in the endosomal Na^+^/H^+^ exchanger NHE9, the interaction of the PI(3,5)P2 lipid with an extended charged loop domain unique to endosomal NHE’s increases its affinity for Na^+^ [28]. Consistently, in vivo, NHE9 is inactive when mistargeted to the plasma membrane but becomes active when localized to endosomes, indicating that the endosomal-specific PI(3,5)P2 lipid is required to activate the protein upon correct trafficking [28]. For NHA2, it has been proposed that the observed lipid-driven remodeling of the dimer interface might be triggered by changes to the cell volume [8]. Whilst plasma membrane NHE1 and NHE3 isoforms have established roles in cell volume regulation [29], this has yet to be established for NHA2. In support of this hypothesis, however, NHA2 was shown to be the most highly differentially expressed protein in the blood group variant Dantu, which is a blood group polymorphism that provides resistance against the most severe forms of malaria [30]. It was shown that the Dantu red blood cell phenotype has an altered cell membrane tension, which decreases the probability of the *Plasmodium falciparum* parasite to successfully invade red blood cells [30]. This protective effect could be reversed by the broad-spectrum inhibitor phloretin, which also inhibits NHA2 [11,30]. Certainly, inhibitors targeting NHA2 would be useful for gaining a better understanding of its physiological role in human [12]. Since the loss of NHA2 reduces blood pressure, NHA2 inhibitors could potentially be developed into therapeutics targeting hypertension [12]. Furthermore, autosomal dominant polycystic kidney disease (ADPKD) is the most frequent inherited kidney disease, and increased NHA2 expression correlates with disease severity [31]. Thus, the inhibition of NHA2 also represents a promising therapeutic target for the treatment of ADPKD [12].

Here, we set out to determine the structure of human NHA2 and to investigate its mechanism of inhibition by the small molecule phloretin.

## 2. Results

### 2.1. Single Particle Cryogenic Electron Microscopy Structure of Human NHA2-Fab Complex

Previously, bison NHA2 was selected for structural studies, as it is more stable following detergent extraction from S. cerevisiae as compared to the human isoform [8]. Nevertheless, a human NHA2 construct lacking the first 58 amino acids (removed due to predicted disorder) rescued growth in the salt-sensitive *S. cerevisiae* AB11c strain [11,32], which lacks the main Na^+^- and K^+^- extrusion systems (Figure 1a). In contrast, poor *S. cerevisiae* growth was apparent under Na^+^ salt stress conditions for either non-induced cells or the double mutant of the ion binding aspartates in human NHA2_∆N_ (D278C, D279C) (Figure 1a). Human NHA2_∆N_ could be purified in a well-folded state, as judged by its size exclusion profile in DDM/CHS (Appendix A). To facilitate image alignment by cryo-EM, a monoclonal antibody was raised in mouse against purified bison NHA2_∆N_, since this homologue is more detergent-stable and still shares high sequence identity (~86%) with human NHA2 (Methods, Appendix A). Mouse hybridoma clones producing antibodies recognizing conformational epitopes in bison NHA2_∆N_ were selected by an enzyme-linked immunosorbent assay on immobilized biotinylated proteoliposomes (liposome ELISA), allowing for the positive selection of the antibodies that recognized the native conformation of NHA2. A monoclonal Fab against bison NHA2_∆N_ was isolated (Methods). Human NHA2_∆N_ was reconstituted with yeast PI lipids into MSP1E2 nanodiscs, which were previously used for the structural investigation of bison NHA2_∆N_ (Methods) [8]. Following reconstitution into nanodiscs, we could confirm that the isolated Fab formed a complex with human NHA2_∆N_, as analyzed by SEC and blue native gel electrophoresis (Appendix A). The NHA2_∆N_-Fab complex peak was, therefore, collected for cryo-EM investigation; hereafter, the NHA2_∆N_ structural construct is referred to as NHA2 only.

After sample preparation, data collection, processing, and refinement, cryo-EM maps of the human NHA2-Fab complex could be obtained to 2.8 Å resolution (FSC = 0.143 criterion) with map density extending to 2.4 Å for many of the TM helices (Appendix A, Appendix A). Human NHA2 retained the expected homodimer with a Fab bound to each protomer (Figure 1b). The majority of the Fab interactions were found to be with residues located on the extracellular surface of NHA2 (Figure 1c). In particular, the charged residues D485, D489, R414, and E416 at the extracellular ends of TM11 and TM13 formed salt bridge and hydrogen bond interactions to residues in a flexible loop of the Fab fragment. Moreover, tyrosine residues Y54, Y102, and Y106 in the Fab formed interactions with charged residues R157, R476, and E484 in the dimer domain (Figure 1c). The extensive extracellular interactions between the Fab fragment and NHA2 indicated that the antibody fragment might inhibit in vivo NHA2 activity. Indeed, Fab addition to growing AB11c yeast cells inhibited the salt stress complementation by human NHA2, at least down to a measured final concentration of ~0.1 μM (Figure 1a).

### 2.2. Homodimer Positioning and Lipid-Mediated Dimerization

Previously, cryo-EM structures have shown that NHA2 is made up from 14-TMs, rather than the 13-TMs observed in the mammalian NHEs [21,33]. The additional N-terminal helix (TM−1) in the dimer domain is domain-swapped, mediating homodimerization by interacting with TM8 and the TM7−TM8 loop from the neighboring protomer (Figure 2a). An unusual feature of the bison NHA2 structure was the highly tilted angle of TM−1 and the length of the connecting loop to TM1, which meant that the NHA2 protomers are held apart by ~25 Å on the intracellular side [8]. Interestingly, this intracellular gap was closed when bison NHA2 was reconstituted into nanodiscs with crude PI lipids. The cryo-EM structure of human NHA2 in nanodiscs is overall very similar to the compacted bison NHA2 structure in nanodiscs with an r.m.s.d. of ~1.2 Å following superimposition (Figure 2b,c).

Oligomerization is thought to be required for Na^+^/H^+^ exchanger activity [20,34] and in elevator proteins in general [24], perhaps by anchoring the scaffold domain [24]. Consistent with this assumption, the deletion of TM − 1 in bison NHA2 became monomeric and was no longer able to alleviate salt stress in the yeast complementation assay [8]. Interestingly, despite the reconstitution of human NHA2 into nanodiscs with crude yeast PI lipids, we observed density for two phosphatidic acid (PA) lipids on the extracellular side of the high-quality cryo-EM maps (Figure 3a). The negatively charged phosphate headgroup and the fatty acid ester linkages are coordinated by two highly conserved lysine residues, K169 and K171, which are located in the scaffold helix TM3. The phosphate headgroup further interacts with the indole nitrogen of the highly conserved tryptophan residue W172, which also forms stacking interactions with the glycerol PA backbone (Figure 3a). We forthwith heated human NHA2 in detergent at 50 °C for 10 min in the presence of either DDM, synthetic, or crude lipids solubilized in DDM (Methods). Indeed, we could confirm that the synthetic PA lipid was clearly the most effective at stabilizing the NHA2 homodimer under these conditions (Figure 3b,c). In the bison NHA2 structure, PI lipids were modelled at the PA lipid location, because native MS had shown that the homodimer retained lipids with a major lipid peak matching the mass of PI, and further, yeast PI lipids were shown to stabilize the homodimer [8]. Nevertheless, the native MS of bison NHA2 had also revealed a small lipid peak at 632 ± 12 Da for the homodimer, which would correspond to the size of a PA lipid [8]. Given that the lipid coordinating residues are conserved between human and bison NHA2 (Appendix A), it is likely that PA would also bind to bison NHA2. Indeed, we re-tested bison NHA2 and found that the PA lipid was highly stabilizing, with a ∆*T_M_* shift of 10 °C as compared to no clear stabilization with synthetic PC lipid addition (Figure 3d).

Because human NHA2 is less detergent-stable than bison NHA2, upon heating, the protein forms higher oligomeric species resembling aggregates (Figure 3b). In contrast, bison NHA2 is more detergent-stable, and the monomeric bison NHA2 population that forms after heating does not aggregate in detergent [8]. After heating bison NHA2, we observed that the dimer: monomer equilibrium shifts to 50:50 (Methods). We increased the final concentration of synthetic PA lipids added. Satisfactorily, we could observe a clear concentration-dependent PA lipid stabilization of the bison NHA2 homodimer (Figure 3e). Whilst we can confirm that PA lipids likely bind to the extracellular dimer interface of NHA2, the identity of the lipids on the cytoplasmic side of the dimer interface is still unclear (Figure 3a). This lipid site stabilized the compacted homodimer form of bison NHA2 [8]. Previously, we had modelled PI lipids in this location, based on native MS, thermostability, and structural analysis [8]. However, the additional lipid density of human NHA2 is ambiguous and, unlike bison NHA2 [8], crude PI lipids were not stabilizing human NHA2 under these conditions (Figure 3c). Further biochemical and structural investigation will be required to confirm the identity of the lipid that binds to dimer interface of NHA2 on the cytoplasmic side and its role in regulating ion exchange activity.

### 2.3. Ion Binding Site in Human NHA2 Reveals New Salt Bridge Interactions

As expected, the human NHA2 structure has a cavity open to the outside that is negatively charged (Figure 4a). The ion binding site in Na^+^/H^+^ exchangers is made up of two half-helices, TM5a-5b and TM12a-12b, which cross-over each other at their breakpoints (Figure 4a). Although the structures of Na^+^/H^+^ exchangers bound to Na^+^ have yet to be determined, the structure of the related K^+^/H^+^ exchanger KefC has been determined with K^+^ binding, which demonstrated how the ion bound to residues located in TM5a-5b and TM6 helices in the core domain [35]. Although residues from the TM12a-12b helices in KefC were not found to coordinate K^+^, TM5a-5b and TM12a-b helices were indirectly connected to each other by salt bridge residues [35]. In human NHA2, we observe that R432 in TM11 forms a salt bridge to E215, which also forms a charged interaction to K460 located in the TM12a-12b breakpoint (Figure 4b). Interestingly, the ion binding aspartate D278 has also formed a salt bridge with R432. Whilst a salt bridge between these residues was predicted by co-evolution analysis [36] in the bison NHA2 structure, R432 formed a π-cation interaction with W457 instead [8]. In the electrogenic bacterial Na^+^/H^+^ exchangers NhaA and NapA, the ion binding aspartate formed a salt bridge interaction with a lysine residue equivalent to R432 [9,37,38]. MD simulations have shown that this salt bridge is broken upon Na^+^ binding [9]. Indeed, the AlphaFold3 [39] prediction of Na^+^ binding to human NHA2 also requires the coordination of D278, and as such, the D278-R432 salt bridge is broken (Figure 4c,d and Appendix A).

Since the bison NHA2 structure matches the AlphaFold 3 prediction of human NHA2 with Na^+^, it seems likely that the bison NHA2 structure may represent a state more likely to bind Na^+^. We propose that Na^+^ binding to NHA2 would catalyze D278-R432 salt bridge breakage and that the unpaired R432 could then fully interact with E215 and W457 residues. Furthermore, the movement of E215 to interact with R432 could break its interaction with K460, as seen in bison NHA2 structure (Figure 4c). Thus, sodium binding to TM5a-5b could trigger a series of electrostatic interactions to alter the conformation of TM12a-12b helices, which would prime the core domain for Na^+^ translocation.

### 2.4. Human NHA2_∆N_ in Complex with the Inhibitor Phloretin

The activity of NHA2 is linked to hypertension and insulin secretion [18,19]. However, the lack of specific tool compounds targeting NHA2 has hindered a deeper understanding of its physiological role. The small molecule phloretin has been shown to inhibit recombinant human NHA2 activity in yeast, and certain mutations in NHA2 weakened this inhibitory effect [11,40]. We could indeed confirm the expected inhibition of human NHA2 by phloretin in the salt-sensitive AB11c strain yeast strain (Figure 5a). Whilst the off-target effects of phloretin make the small molecule unsuitable as a tool compound, understanding how phloretin inhibits NHA2 could, nonetheless, prove useful for the rationale-based design of more effective and selective NHA2 inhibitors. Moreover, confirming a specific interaction of NHA2 with phloretin would help to establish its importance in the previously reported red blood cell tension phenotype [30]. Subsequently, human NHA2-Fab in nanodiscs was prepared, but with the addition of phloretin to a final concentration of 1 mM prior to grid preparation and subsequent cryo-EM studies (Methods). The resulting cryo-EM maps of the NHA2-Fab complex with phloretin reached an estimated resolution of 2.9 Å (FSC = 0.143 criterion) (Appendix A, Appendix A). The overall structures of the NHA2-Fab complex with and without phloretin are very similar, including the presence of PA lipids at the extracellular dimer interface (Figure 5b). Additional map density for phloretin was observed in the extracellular-facing cavity, located between the core and dimer domains (Appendix A).

Phloretin mainly forms hydrophobic contacts, with some polar contributions from D279 and G361 (Figure 5b,c). Nevertheless, the lack of discerning polar interactions and the symmetric shape of phloretin made the discrimination of the exact orientation challenging (Appendix A). The phloretin pose was modelled based on the strongest map density, but the presence of an additional non-modelled density highlights the flexibility of phloretin binding at the rather open and solvent exposed binding position. Interestingly, the sidechain of the strictly conserved ion binding aspartate D279 was repositioned by the presence of the 2,4,6,-trihydroxyphenol moiety of phloretin and was observed within hydrogen bonding distance to the C4-hydroxyl group. The second polar interaction was to the main chain amine of G361. Moreover, conserved hydrophobic residues, L181 in TM3 (scaffold domain) and I283 in TM6 (core domain), both interact with phloretin, and these residues are important for gating an ion-bound state [8,21].

Previously, it was shown that the mutation of P246 to threonine abolished phloretin inhibition [40]. Whilst P246 is more than ~5 Å from phloretin, this residue is located on the TM5a-TM5b peptide break, and so it is possible that the P246T mutant could influence the positioning of TM5b. In contrast, an S178 to threonine mutation was found to be more sensitive to inhibition by phloretin [40]. While the side chain of S178 is positioned almost ~6 Å from phloretin, the longer sidechain of threonine could potentially interact. To corroborate previous analysis [40], we re-screened human NHA2 mutations in the salt-sensitive yeast strain AB11c and indeed found that P246S/T variants displayed weakened inhibition by phloretin (Appendix A). However, S178A/F variants showed poor complementation, rather than their previously reported complementation [40], which could be the consequence of using different salt-sensitive yeast strains. As such, it is unclear if the reported increased sensitivity to phloretin [40] is because of a direct interaction with the small molecule and/or because the S178 mutations are less active in general. Variants L181A/F and P360A/S/T also showed poor complementation (Appendix A). The L181S/T showed partial complementation but were still inhibited by phloretin (Appendix A). Taken together, due to the sensitivity of the outward-facing cavity to mutagenesis, it was difficult to further assess the recognition for phloretin using in vivo complementation assays.

## 3. Discussion

The Fab fragment has facilitated the attainment of high quality cryo-EM maps of human NHA2. Comparing the herein described human NHA2 structure with the previous structure of bison NHA2, as well as the AlphaFold3 [39] modelling of human NHA2 with Na^+^, has enabled a deeper insight into how Na^+^ binding could catalyze core domain movement, by triggering the rearrangements of electrostatic interactions—a fundamental question that has been difficult to ascertain without either ion-bound structures or intermediate states. Certainly, the map quality for the previous bison NHA2 structure was sufficient to assign the positions of sidechains within the ion binding site [8]. Moreover, the broken D278-R432 salt bridge observed in the bison NHA2 matches the AF3 prediction of human NHA2 with Na^+^. Taken together, this is the first example where we have been able to determine different ion binding site configurations for the same major (outward/inward-facing) conformation in a Na^+^/H^+^ exchanger. It is unclear whether the ability to capture intermediates of the NHA2 ion binding site is due to the stabilization by the Fab fragment and/or because of the evolutionarily divergent ion binding site of NHA2, which has additional charged interactions [8,36].

In the human NHA2-Fab structure, we could further unambiguously assign PA lipids bound at the dimer domain interface on the extracellular side. We think it is likely that these PA lipids were mis-assigned as a PI lipids at this location in the previous bison NHA2 structure [8]. We find it surprising that this very minor lipid (>1–2% of total lipids) is able to be retained during purification in detergent as well as during reconstitution into nanodiscs in the presence of high concentrations of other, crude PI lipids. From a technical viewpoint, this example highlights the challenges of assigning lipids in cryo-EM structures and assessing their functional impact to membrane protein function. Notwithstanding, there is likely a sliding scale of lipid affinities and preferences [41], and so some protein surfaces can likely bind to several different lipids, i.e., so the lipid finally refined in the cryo-EM maps is likely to be further influenced by how the protein was purified. With this caveat in place, from a biological perspective, we find it interesting that the minor PA lipid was also found to be bound to the extracellular dimerization interface of the sperm-specific Na^+^/H^+^ exchanger SLC9C1 [8], and the PA lipid also stabilized a more compacted homodimer interface, as compared to the SLC9C1 structure without PA bound. Red blood cells are textbook examples of lipid bilayer asymmetry [42], and although PA is a very minor lipid (making its bilayer asymmetry more difficult to assess), the predominant residence side of PA is expected to be the inner leaflet. Notably, the scaffold helices of the dimer domain in NHA2 are very short, and their ability to thin membranes would be possible, as seen in the bacterial homologue NapA [43]. It, therefore, might be sufficient for the dimerization of NHA2 in of itself to catalyze lipid flipping, but only PA is retained, due to positively charged residues located at the dimerization interface. Furthermore, the PA headgroup has a pKa value close to physiological pH, and for this reason, the PA lipid has been referred to as a pH sensor lipid [44]. The binding of a pH sensitive lipid to Na^+^/H^+^ exchangers regulating intracellular pH seems to be more than a coincidence.

Unfortunately, there are currently no assays for monitoring the flipping of PA lipids. Furthermore, the likely PI-containing lipid on the cytoplasmic side is also required for the larger TM − 1 rearrangement, as seen in bison NHA2 [8]. How the extracellular PA lipid site and the cytoplasmic PI coordinating lipid sites work together is unclear. It is also possible that these lipid interactions can differ in a tissue specific manner, since NHA2 can localize to either organelles or the plasma membrane [12]. Clearly, in vivo imaging will be required to establish the fraction of NHA2 monomers versus homodimers in different cell types and if, or how, these populations change with respect to trafficking and/or cell volume. Whilst it is clear that more extensive investigation is needed, our cryo-EM structure of human NHA2, nonetheless, supports the growing evidence that the preservation of lipid bilayer asymmetry could be more than just to regulate larger signaling pathways, like apoptosis, but lipid flipping could also catalyze more nuanced, protein-specific, cellular processes [45].

Lastly, the structure of human NHA2 in complex with phloretin confirms its cell-based inhibition. While the effect of phloretin on red blood cell tension cannot be fully attributed to the activity of NHA2, our analysis is consistent with its proposed function in the Dantu blood group [30], since the cell tension phenotype could be reversed by increasing concentrations of phloretin [30]. Na^+^/H^+^ exchangers are well-established drug targets, and some small molecules have progressed to clinical trials [46,47,48]. There are some structures presented, such as the cryo-EM structure of human NHE1 in complex with the small molecule inhibitor cariporide (PDB: 7DSX) [22]. Similar to phloretin, this amiloride derivative bound in the outward-facing state between the core and dimer domains, also with a direct electrostatic interaction to the conserved ion binding aspartate D267 [22]. Both cariporide and phloretin are fairly non-specific inhibitors and yet seem to have a similar mode of inhibition, by wedging themselves between the core and dimer domains and thus restricting Na^+^ binding and/or transport. Interestingly, it was recently shown that a similar-sized small molecule also inhibited the elevator sodium–dicarboxylate cotransporter (NaDC3) by binding between core and dimer domain surfaces [49]. Moreover, the pyridine ring of the inhibitor was observed with two different binding positions, similar to that seen herein for phloretin binding to human NHA2. The successful sodium-glucose co-transporter SGLT2 inhibitors used for the treatment of type 2 diabetes illustrate how specific SLCs inhibitors can be developed by extending the molecule from the substrate-binding pocket to interact with gating regions to inhibit transport [50]. It, thus, seems that extending these small molecules beyond these observed binding poses could prove a viable strategy for the development of more specific inhibitors against NHA2 and SLC transporters in general. Moreover, although antibody fragments against intracellular targets are not optimal for technical reasons, it is possible that a synthetic peptide could be engineered to mimic how the Fab binds to human NHA2. Notwithstanding, it will be possible to use the Fab fragment to probe the activity of NHA2 in cells, where the protein is expressed at the plasma membrane.

In conclusion, our structural analysis of human NHA2 not only provides new insights into ion coupling and lipid-mediated regulation but also facilitates the development of tool compounds for better assessing the role of NHA2 in human physiology as well as its potential as a drug target for the benefit of human health and disease.

## 4. Materials and Methods

### 4.1. Human NHA2_∆N_ and Variants Expression and Purification

Homo sapiens NHA2_∆N_ (residues 59–537) and its variants were synthesized and subcloned into the pDDGFP3 vector, which contains a GAL1-inducible promoter, a TEV cleavage site, and N-terminal GFP-TwinStrep-His_8_ tags. The construct was transformed into *Saccharomyces cerevisiae* strain FGY217 following previously established protocols [51,52]. Protein expression was carried out in selective media by induction with 2% (*w*/*v*) galactose. After 22 h of induction, cells were harvested, resuspended in lysis buffer (50 mM Tris-HCl pH 8.0, 1 mM EDTA, 0.6 M sorbitol), and disrupted mechanically, as previously described [51,52]. Membrane was isolated through differential centrifugation and homogenized in buffer containing 20 mM Tris-HCl (pH 7.5) and 0.3 M sucrose, as previously described [51,52]. The solubilization of membrane proteins was performed using 1% (*w*/*v*) DDM and 0.2% (*w*/*v*) CHS, followed by ultracentrifugation to remove insoluble material. The supernatant was incubated with Ni-NTA resin pre-equilibrated in wash buffer containing mild detergent and imidazole. After extensive washing, bound proteins were eluted in imidazole-containing buffer and further purified by size-exclusion chromatography using a buffer containing 20 mM Tris-HCl (pH 7.5), 150 mM NaCl, 0.03% DDM, and 0.006% CHS, as previously described [8].

### 4.2. Cryo-EM Sample Preparation

Cryo-EM samples were prepared following procedures previously established in our earlier study [8], with slight modifications. SEC-purified human NHA2_∆N_-GFP fusion protein in a buffer containing 0.03% DDM and 0.006% CHS was first incubated with Fab fragments at a 1:2 molar ratio. The Fab-bound complex was then mixed with MSP1E2, from which the His_6_-tag had been previously removed, along with yeast phosphatidylinositol lipids (Lardon, Solna, Sweden) at a molar ratio of 1:5:50, respectively, and the mixture was incubated on ice for 30 min. Detergent was removed by incubation with pre-equilibrated Bio-Beads overnight at 4 °C. After the removal of the beads by centrifugation, the supernatant was incubated in batch with Ni-NTA resin to remove empty nanodiscs. Unbound material was washed, and nanodisc-incorporated protein was eluted with imidazole-containing buffer. Protein concentration was quantified via GFP fluorescence, and TEV protease was added at a 2:1 (*w*/*w*) ratio. The sample was dialyzed overnight, concentrated, and subjected to a final round of SEC. The peak corresponding to the nanodisc-reconstituted NHA2-Fab complex was collected and concentrated to 4 mg/mL for cryo-EM. To generate the phloretin-bound sample, the purified complex was incubated with phloretin (Sigma-Aldrich, St. Louis, MO, USA) at a final concentration of 1 mM on ice for 30 min prior to grid preparation.

### 4.3. Determination of Lipid Preferences of NHA2_ΔΝ_ by GFP-Based Thermal Shift Assay

The characterization of thermostability and the lipid thermal stabilization of NHA2 were determined, as previously described [8,53,54]. Purified samples were diluted to a final concentration of ~0.05  mg/mL in buffer containing 20  mM Tris-HCl pH  8.0, 150  mM NaCl, 1% (*w*/*v*) DDM and split into triplicates of 100 µL each. Stock solutions of the respective lipids, 1-palmitoyl-2-oleoyl-sn-glycero-3-phosphate (Avanti research, Birmingham, AL, USA), Phosphatidylinositol (Lardon, Solna, Sweden), and 1-palmitoyl-2-oleoyl-sn-glycero-3-phosphocholine (Avanti research, Birmingham, AL, USA) were prepared by solubilization in 10% DDM at 10 mg/mL. Lipids were added to a final concentration of 1 mg/mL, with a subsequent addition of 1% (*w*/*v*) β-D-octylglucoside (Anatrace, Maumee, OH, USA). The lipid-dependent dimer retention was measured using the FSEC-TS protocol in which the samples were heated at a 50 °C for 10 min and larger aggregates pelleted by centrifugation. A total of 50 µL of the resulting supernatant from each sample was injected onto an Enrich SEC 650 10 × 300 Column (BioRad, Berkeley, CA, USA) preequilibrated in buffer containing 20 mM Tris pH 7.5, 150 mM NaCl, and 0.03% (*w*/*v*) DDM for fluorescence detection size-exclusion chromatography using a Shimadzu HPLC LC-20AD/RF-20A system (Shimadzu Corporation, Kyoto, Japan). The heights of the respective lipid sample dimer peaks were normalized to the of NHA2 added only buffer eluting at 11.5 mL at 5 °C. These intensities were determined from three independent FSEC experiments and plotted. The GFP-TS assay for bison NHA2_ΔΝ_ was performed following the experimental procedures described in previous studies [8]. The apparent melting temperature (*T*_M_) of bison NHA2_ΔΝ_ was calculated by plotting the mean GFP fluorescence intensity from three technical repeats per temperature and fitting it to a sigmoidal logistic regression using Graphpad Prism (10) software. The PA dependent stabilization of bison NHA2_ΔN_ by FSEC-TS was performed at 53  °C (*T_M_*  +  5  °C), as previous described [8].

### 4.4. Generation of the Fab

Bison NHA2_ΔN_ was expressed in *S. cerevisiae* and purified, as previously described [8]. Mouse monoclonal antibodies against bison NHA2 were raised, as follows. A proteoliposome antigen was prepared by reconstituting the purified bison NHA2 protein at high density into phospholipid vesicles consisting of a 10:1 mixture of chicken egg yolk phosphatidylcholine (Avanti research, Birmingham, AL, USA) and the adjuvant lipid A (Sigma-Aldrich, St. Louis, MO, USA) to facilitate immune response. Two female MRL/lpr mice were immunized with the proteoliposome antigen by three injections that were given at two-week intervals. Antibody-producing hybridoma cell lines were generated using a conventional fusion protocol. Biotinylated proteoliposomes were prepared by reconstituting bison NHA2_ΔN_ with a mixture of egg PC and 1,2-dipal-mitoyl-sn-glycero-3-phosphoethanolamine-N-(cap biotinyl) (Avanti research, Birmingham, AL, USA) and used as a binding target for conformation-specific antibody selection. The targets were immobilized onto streptavidin-coated microplates (Nunc, Roskilde, Denmark). Hybridoma clones producing antibodies recognizing conformational epitopes in bison NHA2_ΔN_ were selected by an enzyme-linked immunosorbent assay on immobilized biotinylated proteoliposomes (liposome ELISA), allowing for the positive selection of the antibodies that recognized native bison NHA2_ΔN_. Further, antibody binding to SDS-denatured bison NHA2_ΔN_ was used for negative selection. Stable complex formation between human NHA2_ΔN_ and each antibody clone was confirmed using size-exclusion chromatography (SEC). A monoclonal antibody, which binds to both bison NHA2_ΔN_ and human NHA2_ΔN_, was selected. The sequence of the Fab was determined via standard 5′-RACE using total RNA isolated from hybridoma cell.

### 4.5. Cryo-EM Grid Sample Preparation and Data Acquisition

Nanodisc-reconstituted protein samples were applied individually onto freshly glow-discharged Quantifoil R2/1 Cu300 mesh grids (Quantifoil Micro Tools GmbH, Germany). Grids were blotted for 3 s under 100% humidity and rapidly vitrified in liquid ethane using a Vitrobot Mark IV system (Thermo Fisher Scientific, Waltham, MA, USA). Cryo-EM data acquisition was performed on a Titan Krios G3i transmission electron microscope operating at 300 kV and equipped with a Gatan BioQuantum energy filter (Gatan, Inc., Pleasanton, CA, USA) and a K3 direct electron detector, running in super-resolution mode with hardware binning. Movies were recorded at a nominal magnification of 130,000× using EPU software v3.2.0.4776 (Thermo Scientific, Waltham, MA, USA), with aberration-free image shift and fringe-free imaging enabled. Additional parameters for data collection, including dose rate, exposure time, pixel size, and defocus range, are provided in Appendix A.

### 4.6. Cryo-EM Data Processing

Image processing for all the datasets was performed using the cyoSPARC software v4.5.3 [55]. Movie frames were aligned using the “Patch motion correction multi” function and the contrast transfer function was estimated using the “patch ctf multi” algorithm. The apo dataset contained 20,005 micrographs, which were motion-corrected using patch motion correction. After CTF correction by Patch CTF estimation, 15,392 micrographs were subjected to the Blob picker for automatic particle picking. Around 6.8 million particles were extracted and subjected to several rounds of 2D classification in cryoSPARC v4 [55]. Around 1.1 million particles, belonging to good 2D classes were selected and subjected to multi model ab initio reconstruction. A good class containing 483,175 particles was selected and subjected to another round of hetero refinement and multi-model ab initio reconstruction to remove particles corresponding to junk classes. Several rounds of 3D classification resulted in 335,205 particles corresponding to the human NHA2-Fab complex. The nonuniform refinement, CTF refinement, reference-based motion correction (RBMC), and nonuniform refinement of NHA2-Fab yielded a final reconstruction at 2.8 Å resolution at gold standard FSC (0.143). To improve the map features for the transporter region, volume corresponding to Fab was masked out and local refinements were performed, which was the gold standard FSC resolution to 2.9 Å.

For the phloretin bound structure, 15,658 out of 16,151 micrographs had an estimated ctf resolution better than 5Å and were selected for further image processing. Around 4.4 million particles were extracted after template-based picking and subjected to multiple rounds of 2D classification. A total of 758,568 particles were selected, and initial 3D volumes were obtained using multi-class ab initio reconstruction. A total of 280,592 particles corresponding to a good 3D reconstruction were further cleaned up using several rounds of hetero-refinement. A total of 251,355 particles were finally selected and refined to high resolution using nonuniform refinement. To improve the map features for the transporter region, the volume corresponding to Fab was masked out, and local refinements were performed. The final reconstruction had an overall resolution of 2.8 Å based on the gold standard FSC at 0.143.

### 4.7. Model Building and Refinement

The predicted AlphaFold3 [39] model of Fab was manually examined and adjusted using Coot [56]. The model of the bison NHA2_ΔΝ_ structure was fitted into the map density using the fit in the map utility of ChimeraX [57]. The model was refined using Phenix [58], and real-space refinement was performed. Model building for the phloretin-bound NHA2 was also performed using the apo structure as a starting model followed by manual adjustment in Coot. The model was refined in Phenix using real space refinement. For the prediction location of the Na^+^ ion binding site, AlphaFold 3 [39] was performed on the human NHA2 homodimer using the online version with default settings and Na^+^ selected as the ligand (https://alphafoldserver.com/welcome (accessed on 28 November 2024)). The resulting models have been deposited in the PDB/EMDM with the following accession codes: human NHA2_∆N_-Fab complex (apo), EMD-53377, and PDB 9QUB; phloretin-bound human NHA2_∆N_-Fab complex, EMD-53384, and PDB 9QUW.

### 4.8. Complementation Assay for Human NHA2 Against Salt Stress in Yeast

*S. cerevisiae* strain AB11c (*ena1-4Δnhx1Δnha1Δ*), which lacks endogenous Na^+^ and Li^+^ efflux transporters [32], was employed for the heterologous expression of human NHA2 constructs, following a previously established bison NHA2 study [8]. Seed cultures were grown to saturation and diluted in selective −URA media containing 2% (*w*/*v*) galactose. For the assay, cultures were adjusted to an initial OD_600_ of 0.2 for Li^+^ or 0.02 for Na^+^ toxicity testing. The various concentrations of either LiCl or NaCl were added, and the final volume was adjusted to 200 μL in a 96-well plate. Plates were incubated at 30 °C for 48–72 h. To test the effect of phloretin on NHA2 activity and its variants, the compound was added to the culture medium at a final concentration of 500 μM (prepared from a 100 mM stock in DMSO). After incubation, cultures were resuspended, and OD_600_ measurements were recorded using a SpectraMax plate reader (Molecular Devices, San Jose, USA). All data shown are representative of at least three independent biological replicates.

## Figures and Tables

**Figure 1 ijms-26-04221-f001:**
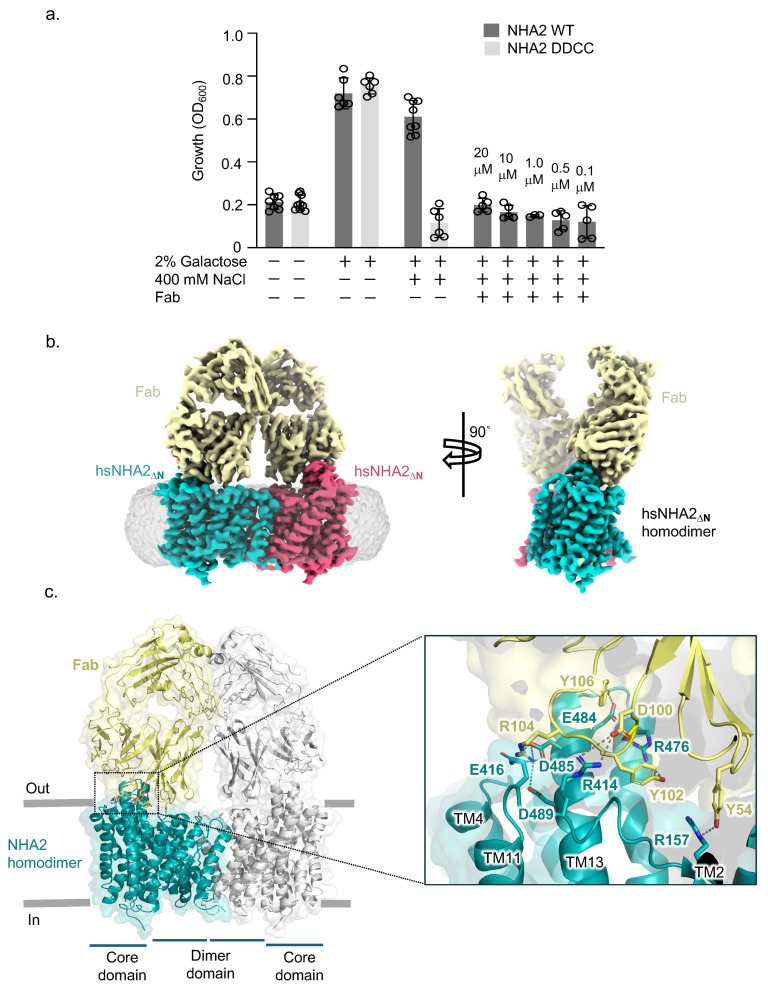
(**a**) Salt tolerance of *S. cerevisiae* AB11c yeast strain with and without the GAL-induced expression of human NHA2_∆N_ and ion binding mutant +/− Fab. (**b**) Cryo-EM map of human NHA2_∆N_-Fab with the maps for the respective protomers colored teal/red and the Fab. The map before micelle subtraction is shown in grey (**left**). (**c**) Structure of hsNHA2_∆N_-Fab dimer, with one hsNHA2_∆N_ monomer: Fab pair colored (teal and yellow) and the second pair shown in grey, with all protein shown as cartoon and selected residues as sticks (**right**). Magnification of the hsNHA2_∆N_-Fab interface, highlighting key interactions between NHA2_∆N_ and Fab (colored as in (**b**)) with residues shown as sticks, bonds indicated by dashed lines, and protein as cartoon.

**Figure 2 ijms-26-04221-f002:**
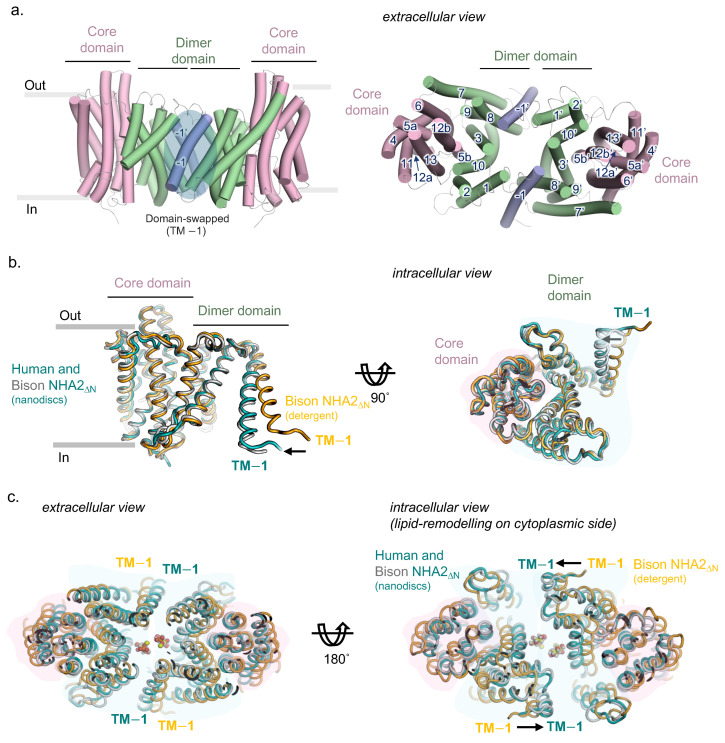
(**a**) Dimeric NHA2 viewed along the membrane plane (**left**) and from the extracellular side (**right**), with helices shown as cartoon. The ion translocation 6-TM core domain (transporting) is colored pink, the dimerization domain green, and the N-terminal domain-swapped transmembrane helices (TM−1) blue, with the respective transmembrane helices enumerated. (**b**) Superimposition of 14-TM monomers of NHA2 structures in two orientations, along the membrane plane (**left**) and from the extracellular side (**right**), showing the different position of the TM−1 of the two nanodisc structures hsNHA2_ΔN_ (teal cartoon) and bison NHA2_ΔN_ (PDB 7P1K, orange cartoon), compared to the bison NHA2_ΔN_ in detergent (PDB 7P1J, grey cartoon), with the respective NHA2 domains shaded colored as in (**a**). (**c**) Homodimer of NHA2 structures with extracellular (**left**) and intracellular views (**right**), with protein and domain shadow colored as in (**c**) and lipids shown as sticks.

**Figure 3 ijms-26-04221-f003:**
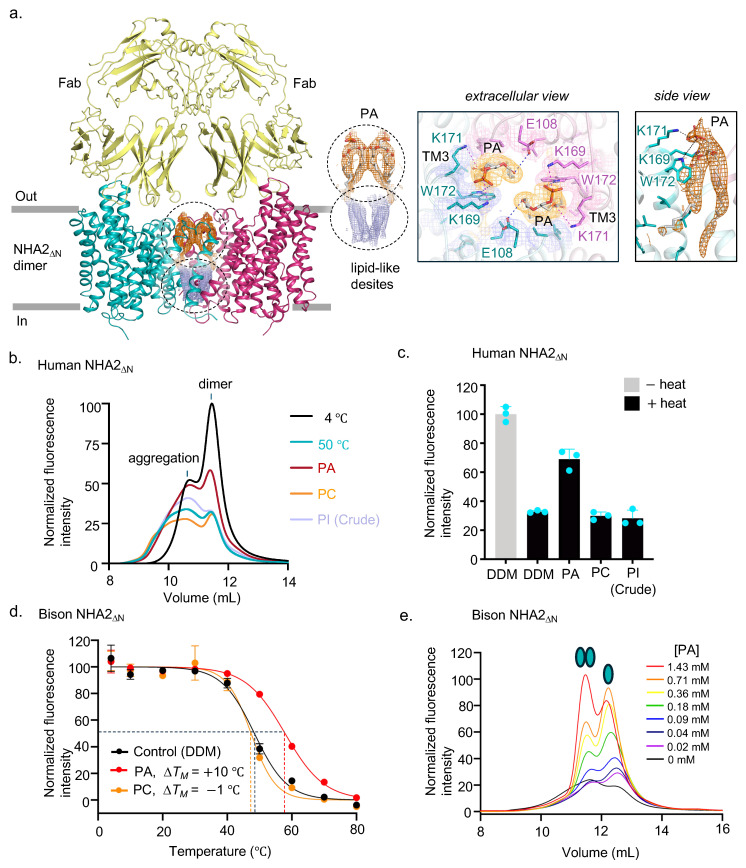
NHA2 PA binding and stabilization. (**a**) Binding position of PA lipids (grey sticks) in the hsNHA2_ΔN_-Fab complex structure (cartoon and with selected residues as sticks, colored as in Figure 1b), with the cryo-EM density maps shown for the PA lipids (orange mesh), protein (colored per domain), and additional lipid-like density (grey mesh). The putative polar interactions are highlighted by dashed lines (magenta and blue). (**b**) Representative FSEC traces of DDM/CHS-purified hsNHA2 _ΔN_ after heating at T_M_ + 5 °C for 10 min in the presence of DDM (black) compared to lipids. (**c**) Thermal stabilization of purified dimeric hsNHA2 _ΔN_-GFP in the presence of DDM compared to lipids. The data are normalized fluorescence, and error bars are the s.d. of n = 3 independent experiments. (**d**) Thermal shift stabilization of purified bison NHA2 _ΔN_-GFP in the presence of DDM (black) compared to PA in DDM (red) and PC in DDM (orange). The data are normalized fluorescence mean  ±  s.e.m. of n  =  3 independent experiments. The apparent melting temperature TM was calculated with a sigmoidal four-parameter logistic regression function (Methods). (**e**) PA-dependent stabilization of bison NHA2 _ΔN_-GFP dimer as assessed by FSEC-TS.

**Figure 4 ijms-26-04221-f004:**
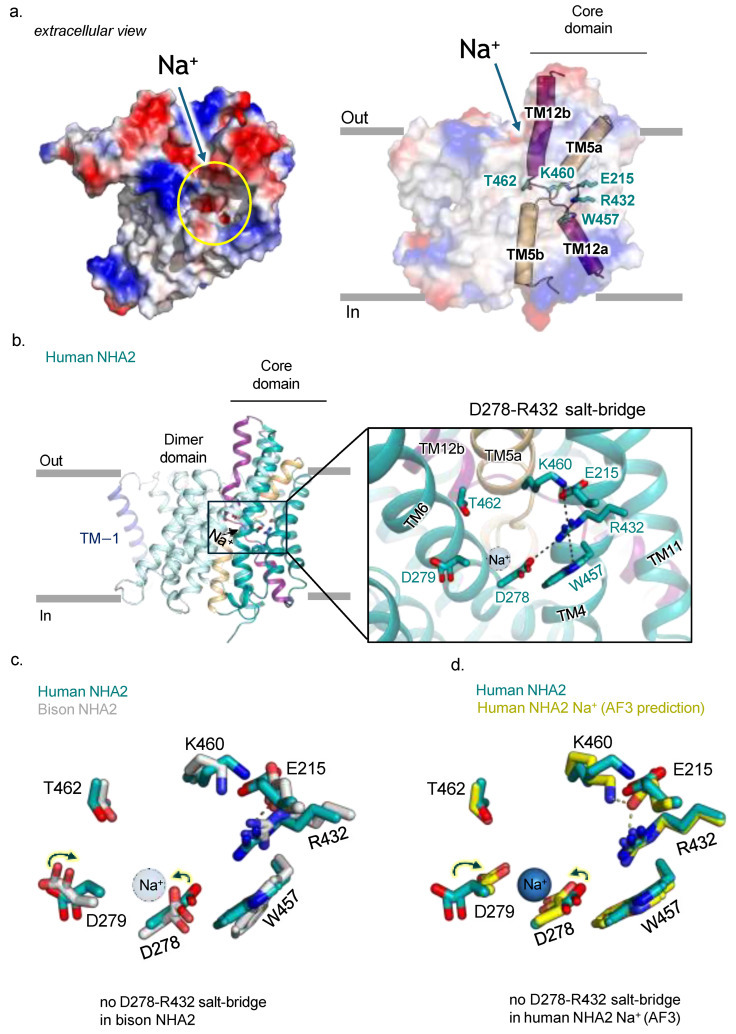
Human NHA2 ion binding site. (**a**) Left: Electrostatic surface representation for the extracellular surface of the hsNHA2_∆N_ outward-facing monomer. The yellow circle highlights the outward-facing cation attracting funnel (left inset). Cartoon representation of hsNHA2_ΔN_ in nanodiscs with the electrostatic surface representation, showing the two pairs of half-helices and key residues (as sticks labelled). The crossover of half helices TM5a-b and TM12a-b (cartoon) are unique to the NhaA-fold. (**b**) The ion binding site predicted by Alphafold3 (AF3). Side view of the hsNHA2 monomer (as cartoon) with the predicted sodium ion shown as a magenta sphere. Right: Magnification of the ion binding site of the hsNHA2_∆N_, which has the two aspartates seen in electrogenic Na^+^/H^+^ antiporters (D278 and D279, sticks) and the NHA2-specific salt bridge between R432 and E215 connected to W456 (residues as stick). The predicted sodium ion position highlighted by the magenta sphere. (**c**) The ion binding site residues (sticks) of hsNHA2 (teal, PDB 9QUB) and biNHA2 (grey, PDB 7p1k), with the position of the sodium ion highlighted by the sky-blue sphere. (**d**) Comparison of the ion binding site residues between the hsNHA2 structure (PDB 9QUB) and the AF3-predicted model of the Na^+^-bound hsNHA2 structure (yellow). Dashed lines represent hydrogen bonding.

**Figure 5 ijms-26-04221-f005:**
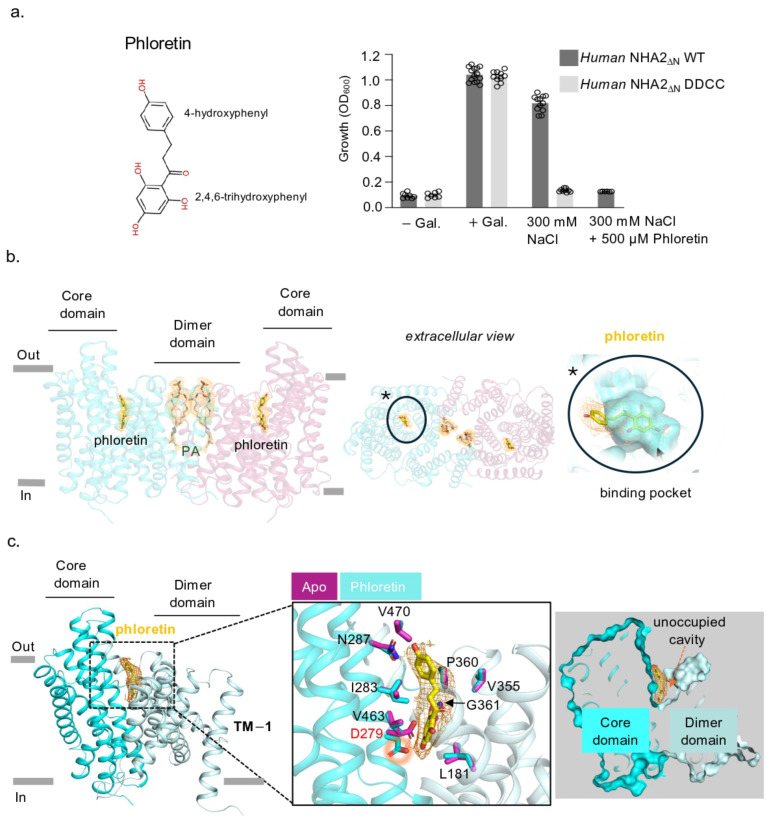
(**a**) Inhibition of NHA2 by phloretin. (**b**) Cryo-EM structure of human NHA2 (cartoon colored as in Figure 1b) with phloretin at the ion binding site, PA at the dimer interface, and the electron density maps for these ligands (yellow mesh), along the membrane (**left**) and from the extracellular side (**center**), and the solvent-exposed pocket (cyan surface), wherein phloretin was bound (**right**); asterisk highlights zoomed in view. (**c**) Binding site of phloretin (yellow sticks) in hsNHA2 (cyan and pale cyan cartoon), with the phloretin density (red mesh). Residues involved in phloretin interaction are highlighted, with apo-state residues shown in purple and phloretin-bound residues shown in cyan. The solvent-exposed and unoccupied cavity directly adjacent to phloretin is viewed in a different angle and indicated by the arrow on the far right.

## Data Availability

Correspondence and request for materials should be addressed to D.D. (ddrew@dbb.su.se).The cryo-EM maps of human NHA2-Fab are available as EMD-53377 and EMD-53384 on the EMDB database, and the corresponding atomic structures are deposited on the Protein Data Bank with accession numbers 9QUB and 9QUW.

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
