# Peer review of "Structure and Inhibition of the Human Na+/H+ Exchanger SLC9B2"

_ijms, 2025, doi:10.3390/ijms26094221_

Round 1
Reviewer 1 Report
Comments and Suggestions for Authors
The manuscript from Jung et al. presents structures of NHA2 (SLC9B2) in complex with Fab fragments. The groups aims to investigate the role of the lipids phosphatic acid (PA) and phosphatidylinositol (PI) in the dimerization and function of human NHA2. Furthermore, the structure of NHA2 in complex with the inhibitor phloretin is shown.
However, the overall quality of the manuscript is poor and needs extensive rewriting, improvement in the presentation of results and figures and to allow for the conclusion that are made from their study. Supplementary figures are depicted but are not presented anywhere in the manuscript. Thus, essential controls to validate their results are missing.
I advise against considering this study for publication in IJMS.
Comments:
Comment 1:
Introduction – Last paragraph: The connection from the open questions to the implications and goals seems lengthy and unfocused. Furthermore, the goal(s) of the study remain(s) short and too vague. It is strongly advised to rewrite the last paragraph to make the implication and goal of the study more focused.
Comment 2:
The supplementary figures are currently not available at this review stage. Please make sure that Fig. S1 to Fig. S5 are available for the review process.
Comment 3:
The resolution of the figures is very poor and details cannot be seen clearly. Therefore, results and conclusions cannot be drawn from the presented figures.
Comment 4:
For the presented structures, the cryo-EM validation data is not presented. I assume they are visible in the supplementary figures.
Comment 5:
Fig. 2e: Axis labels are missing.
Comment 6:
Material and Methods – Section 4.1: Please visualize the purity of the used proteins after purification, for instance by SDS-PAGE and silver staining or corresponding methods.
Comment 7:
Material and Methods – Section 4.3: Please show representative chromatograms.
Comments on the Quality of English LanguageEnglish must be improved. Check for typos, too.
Reviewer 2 Report
Comments and Suggestions for Authors
In their work, Jung and coauthors obtain cryo-electron microscopy structures of human NHA2 in complex with a Fab and with and without the inhibitor phloretin bound at 2.8 to 2.9 Å resolution. The complex with Fab-fragments has enabled cryo-EM maps at a higher resolution compared to previous cryo-EM structures of bison NHA2. Using Fab-fragments to improve cryo-EM resolution is a well-established technique in structural biology.
Authors used a salt-sensitive yeast strain (lacking genes for Na+ (Li+) efflux proteins) to overexpress the hNHA2. The expression of hNHA2 restored the growth of a host cell under high-salt conditions, which served as an effective model for functionality tests (inhibition experiments). In that experimental system, authors assessed the inhibitory effect of phloretin on hsNHA2 activity and its mutants. The growth of the cells was decreased in the presence of phloretin. The authors presented an exploratory data analysis where they visualized patterns (without resorting to statistical analysis). This is a valid experimental approach to show the inhibitory effect. Nevertheless, an assessment of the affinity of phloretin would be a valuable addition to the current work. (The authors used 500 µM of the drug in their experiments.)
The authors produce a cryo-EM structure of an hNHA2 with phloretin bound to it. The structure captured hNHA2 with phloretin wedging between the core and dimer domains, thereby restricting Na+ binding and/or transport. Interestingly, phloretin was shown to have a direct electrostatic interaction with the conserved ion-binding residue, i.e., the drug is dual-targeting.
Authors also attempted to answer the fundamental question of how Na+ binding could catalyze the core domain movement of NHA2. However, structures of Na+/H+ exchangers bound to Na+ are yet to be determined. Therefore, the obtained cryo-EM map of hNHA2 and Alphafold3 were used to predict the sodium binding site in hNHA2. The prediction confirmed that, in the presence of Na+, the salt bridge of an ion-binding residue is disrupted. This phenomenon was also observed in literature in molecular dynamics (MD) simulations. The predicted structure also revealed new salt-bridge interactions for ion-binding residues. Authors speculate that sodium binding to the TM5a-5b domain could trigger a change in the conformation of TM12a-12b, which primes the translocation of the core domain for undergoing elevator transitions. This is a valid, feasible mechanism of elevator transport in hNHA2.
In their cryo-EM structures, authors also observed the unambiguous assignment of phosphatidic acids (PA) at the extracellular surface of the human NHA2 dimer interface. PA is a minor lipid of the plasma membrane. It has a negatively charged headgroup (phosphate). Therefore, its predominant residence is expected to be in the inner leaflet of a membrane. The PA lipid binding at the extracellular dimer interface in NHA2 was unexpected. The authors propose the flipping of PA from the inner leaflet.
The cryo-EM structures of the NHA2 presented in this work represent a novel finding, shedding light on structural and functional mechanisms of the NHA2 exchanger.
Minor points
- Page 4, line 110. “Previously, previously…”
- Page 8, line 231: “…made the discrimination of the exaction orientation challenging”. The word exaction may not be used correctly.
Reviewer 3 Report
Comments and Suggestions for Authors
In this manuscript, the authors report the structure determination of human NHA2 (SLC9B2), a protein whose structure was previously unknown. The authors extensively compare the structure of the human protein with a previously resolved structure of bison NHA2, with very high sequence similarity (93% as reported by the authors), from the same laboratory. The structure of human NHA2 looks essentially identical to the previously resolved bison NHA2 structure in nanodiscs based on the figures, even though the authors do not report RMSD values. The authors first solubilized the protein in detergent solution, then used the same reconstitution method for human NHA2 as they had used for bison NHA2 in their previous study.
Despite the similarity with the previous bison NHA2 structure, the work is still significant, on one hand due to the human protein used, on the other hand due to the structure solved in complex with phloretin, the only available, albeit non-selective, inhibitor for NHA2.
An important topic of the study is lipid binding and lipid-mediated dimerization of NHA2, which has also been observed for the previous bison NHA2 structure. At the interface of the homodimer NHA2, several lipids molecules can be observed in both human and bison structures. The lipids on the cytoplasmic leaflet side have previously been modeled as PI (phosphatidyl inositol) lipids. Here, the authors report being able to unambiguously identify the lipids at the analogous posision as PA (phosphatidic acid) species. They also note that the observed binding site is completely conserved between human and bison NHA2 orthologs. It is curious where the discrepancy between the lipid species in the two structures comes from, since the preparation of the nanodiscs for cryo-EM seemed to be the same in both studies. Difference in amino acid sequence also cannot account for a possible difference in affinity towards PI and PA lipids at this site.
FSEC curves reported by the authors are difficult to compare with those for bison NHA2 (ref. 22) by the same group, since they use a different temperature, and at 50 C, which seems to be over the melting temperature NHA2, the peak corresponding to the monomeric protein is gone, or at least not labeled in Figure 2e (left panel). Why the authors used 50 C as their target temperature instead of the previous 40 C is curious.
It is also remarkable that despite the high sequence similarity and the same preparation steps, the authors seem to lose all fluorescence in the thermal shift assay (Figure 2e, right panel) at 50 C, while the bison NHA2 construct still retained around 40% fluorescence at this temperature (see Figure 3c in ref. 22). The stabilizing effect of PI as seen with bison NHA2 also cannot be recovered using the human protein, even though a yeast PI preparation was used in the latter instead of synthetic PI. Therefore even though the human and bison proteins seem to have different properties, comparison is difficult due to these differences in the experimental protocols.
Overall, based on the discussion in this manuscript, I get the impression that the bound lipids in the previous, bison NHA2 structure might have been misidentified. In fact, the authors themselves state on page 5, line 152 that "Given the lipid-binding site residues are conserved between human and bison NHA2 (Fig. S2), it is likely that PA is also the preferred lipid in both isoforms." The authors also argue in favor of PA being a more suitable regulator of protein function due to its pKa value, and this function is likely conserved between bison and human. If the PI lipid in the bison NHA2 structure is believed to be misassigned, the authors might want to issue an erratum to the previously published paper, or a correction to the files deposited in the PDB. Otherwise, if the authors believe the presence of PI in bison NHA2 was an artefact of the reconstitution process, this should be stated more clearly, but then it should also be explained why the PA assigment in the human NHA2 structure is not a similar artefact. A comparative analysis of the human and bison constructs using the same experimental protocols should be useful to track down the source of this discrepancy.
The overall quality of the manuscript is good, but the resolution of the figures needs to be improved, as in many cases it is on the verge of readability. While the language is overall well readable, a thorough proofreading would be necessary to correct typos and minor grammatical errors in the text.
Round 2
Reviewer 1 Report
Comments and Suggestions for Authors
The authors have provided a significant improvement of the manuscript with high-resolution figures and included the supplementary figures that were previously missing. However, the overall quality of the manuscript can still be significantly improved. Below are further questions I would like to address the authors.
Comment 1:
The figures and supplementary figures contain several artifacts, especially boxes.
Comment 2:
Fig. S1a: For the SDS-PAGE images, I suggest to show full gel images like in Fig. S1c. The Coomassie staining shows a double band, which is not visible in the in-gel fluorescence image.
Comment 3:
Fig. S1a: The Coomassie staining shows many weak bands at lower molecular weight, besides a rather strong one at around 18 kDa. However, the chromatogram would suggest a highly pure protein.
Comment 4:
Section 2.2 – lines 177-180: “The negatively charged phosphate headgroup and the fatty acid ester linkages are coordinated by two highly conserved lysine residues, K169 and W172, which are located in the scaffold helix TM3. Further, the glycerol backbone is interacting with a highly conserved tryptophan residue, K171 (Fig. 3a). ” Lysine and tryptophan residues are mixed up. Please correct.
Comment 5:
Fig. 3a highlights an interaction between the phosphatic acid and E108. However, it is not discussed in the results section.
Comment 6:
Bison NHA2 was treated with the addition of OG and heating, which destabilize the protein. I am wondering whether the structural integrity of bison NHA2 is maintained. Does the destabilized NHA2 promote or facilitate PA binding, which is absent in untreated bison NHA2 or human NHA2?
Comment 7:
It is unclear how the structure of human NHA2 in the presence of sodium was obtained by AlphaFold. Did the authors introduce a sodium ion into the structure? And if so, how was this performed? Further information is also not provided in the methods section.
Comment 8:
“To corroborate previous analysis [40], we re-screened human NHA2 mutations in the salt-sensitive yeast strain AB11c, and indeed found that P246S/T variants were no longer inhibited by phloretin (Fig. S6).”
There is still some inhibition, as indicated by the bars in Fig. S6.
Comment 9:
“Variants L181A/F and P360A/S/T also showed poor complementation (Fig. S6), and although L181S/T showed partial complementation, these variants were still inhibited by phloretin (Fig. S6).”
The growth of bacteria containing the mutants seems independent of the presence of phloretin, since the OD600 is basically the same. Where does the conclusion “still inhibited” come from?
